# Habitual Dietary Fiber Intake, Fecal Microbiota, and Hemoglobin A1c Level in Chinese Patients with Type 2 Diabetes

**DOI:** 10.3390/nu14051003

**Published:** 2022-02-27

**Authors:** Jiongxing Fu, Kelin Xu, Xumin Ni, Xiaoqiang Li, Xiaofeng Zhu, Wanghong Xu

**Affiliations:** 1School of Public Health, Fudan University, 138 Yi Xue Yuan Road, Shanghai 200032, China; 20211020001@fudan.edu.cn (J.F.); xukelin@fudan.edu.cn (K.X.); 16211020003@fudan.edu.cn (X.L.); 2Yiwu Research Institite, Fudan University, Building V of Zhongfu Square, Yiwu 322000, China; 3Department of Population and Quantitative Health Sciences, School of Medicine, Case Western Reserve University, Cleveland, OH 44106, USA; xmni@bjtu.edu.cn (X.N.); xxz10@case.edu (X.Z.); 4Department of Mathematics, School of Science, Beijing Jiaotong University, Beijing 100044, China

**Keywords:** dietary fiber intake, gut microbiota, type 2 diabetes mellitus, hemoglobin A1c

## Abstract

High-fiber diet interventions have been proven to be beneficial for gut microbiota and glycemic control in diabetes patients. However, the effect of a low level of fiber in habitual diets remains unclear. This study aims to examine the associations of habitual dietary fiber intake with gut microbiome profiles among Chinese diabetes patients and identify differential taxa that mediated associations of dietary fiber with HbA1c level. Two cross-sectional studies and one longitudinal study were designed based on two follow-up surveys in a randomized trial conducted during 2015–2017. The study included 356 and 310 participants in the first and second follow-ups, respectively, with 293 participants in common in both surveys. Dietary fiber intake was calculated based on a 3-day 24-h diet recall at each survey and was classified into a lower or a higher group according to the levels taken based on the two surveys using 7.2 g/day as a cut-off value. HbA1c was assayed to assess glycemic status using a cut-off point of 7.0% and 8.0%. Microbiome was profiled by 16S rRNA sequencing. A high habitual dietary fiber intake was associated with a decrease in α-diversity in both the cross-sectional and longitudinal analyses. At the first follow–up, phylum *Firmicutes* and *Fusobacteria* were negatively associated with a higher dietary fiber intake (*p* < 0.05, *Q* < 0.15); at the second follow-up, genus *Adlercreutzia*, *Prevotella*, *Ruminococcus*, and *Desulfovibrio* were less abundant in patients taking higher dietary fiber (*p* < 0.05, *Q* < 0.15); genus *Desulfovibrio* and *Ruminococcaceae (Unknown*), two identified differential taxa by HbA1c level, were negatively associated with dietary fiber intake in both the cross-sectional and longitudinal analyses, and mediated the dietary fiber-HbA1c associations among patients taking dietary fiber ≥ 7.2 g/day (mediation effect β [95%CI]: −0.019 [−0.043, −0.003], *p* = 0.018 and −0.019 [−0.046, −0.003], *p* = 0.016). Our results suggest that habitual dietary fiber intake has a beneficial effect on gut microbiota in Chinese diabetes patients. Further studies are needed to confirm our results.

## 1. Introduction

Type 2 diabetes mellitus (T2DM) has imposed a huge burden on health systems in China due to its high prevalence and rapid upward trend, as well as the epidemic of related complications in diabetes patients [1,2,3]. Effective interventions are urgently needed to prevent and control the metabolic disorder in the Chinese population. In recent years, the role of gut microbiota in the development of T2DM and other metabolic disorders has aroused great attention globally [4,5]. It is estimated that there are 10 to 100 trillion microbes in a human being, and each person harbors more than 1000 phylotypes, mostly located in the large intestine and dominated by phylum *Firmicutes* and *Bacteroidetes* [6,7]. Although no specific single or small group of bacteria has been found fully responsible for diabetes and related diseases yet, the increased conditional pathogenic bacteria and reduced beneficial bacteria have been widely recognized as typical microbial changes for T2DM [4]. Altering gut microbiota has been considered as an intervention tool with a great potential to improve glycemic control in diabetes patients [8].

Gut microbiota could be modulated by soluble and insoluble dietary fiber through providing nutritional substrates and promoting the growth of beneficial bacteria [9,10], including short-chain fatty acids (SCFAs) producing strains [11]. The altered gut microbiota may improve the nutritional status of hosts, reduce insulin resistance, and relieve persistent inflammation, and thus improve the prognosis of diabetes and decrease the risk of complications [12,13,14]. For example, SCFAs produced in the gut are important fuels for intestinal epithelial cells to regulate their barrier, metabolism and immunity functions [12]. Multiple intervention trials have proven the benefits of high-fiber diets in glycemic control among T2DM patients [15,16,17]. Zhao et al. [16] found that adopting a high-fiber diet selectively promoted the growth of SCFA-producing organisms and alleviated diabetes in Chinese patients. A well-known SCFA-producing bacteria is *Bifidobacterium* spp., which has been consistently observed to be notably increased by a high-fiber diet in studies globally [15,16].

A high-fiber diet has been widely applied to improve the glycemic status of diabetes patients [17]. In these intervention trials, diets were specially designed to contain a much higher level of dietary fiber than that found in daily diets. In real practice, however, many diabetes patients are unable to take such a high level of dietary fiber because of eating patterns or poor physical conditions [18], particularly among populations who rarely consume whole grains [19]. We called such daily diets without any supplement interventions habitual diets. It is unclear whether the low-level of fiber in habitual diets could regulate gut microbiota, and if so, what level of dietary fiber would improve glycemic control status in diabetes patients.

In this study, we explored the potential effect of habitual dietary fiber intake using data collected in a randomized controlled trial (RCT) addressing health literacy and/or exercise interventions in Chinese T2DM patients. Specifically, we evaluated associations of habitual dietary fiber intake and hemoglobin A1c (HbA1c) level with diversity of gut microbiota, and identified common differential taxa associated with dietary fiber and HbA1c in cross-sectional and longitudinal studies based on the first and second follow-up surveys of the parent RCT. Our results may help to understand whether and how diabetes patients benefit from habitual dietary fiber intake.

## 2. Materials and Methods

### 2.1. Study Design and Participants

In our previous cluster RCT (number: ISRCTN76130594, registration date: 12 January 2015), a total of 799 Chinese patients with T2DM were recruited from the Changning and Minhang Districts of Shanghai, China, and randomized into four arms to or not to receive health literacy and/or exercise-focused interventions [20], as illustrated in Appendix A. All participants met the following criteria before interventions: (1) diagnosed with T2DM according to the 1999 WHO criteria; (2) were between 18 and 85 years old; (3) most recent HbA1c ≥ 7.5% or fasting plasma glucose (FPG) level ≥ 10 mmol/L; (4) agreed to participate for 2 years; and (5) permitted by their physicians in charge. Exclusion criteria included: (1) low vision (worse than 0.1/4.0 using the Standard Logarithmic Visual Acuity Chart); (2) significant dementia or mental disorder (based on health providers’ report or chart review); (3) terminal illness with anticipated life expectancy < 2 years. All experimental protocols were approved by the Ethics Committee of Fudan University (IRB00002408 & FWA00002399) (approval number: 2013-06-0451), and written informed consent was obtained from all participants prior to their enrollment in the study.

In this analysis, we only included 400 patients from the Minhang District of Shanghai who donated stool samples at the first or the second follow-up surveys. As shown in Figure 1, we designed two cross-sectional studies and a longitudinal study based on the two follow-up surveys conducted at the end of 1-year interventions (from January to March 2016) and 1-year post-interventions (from January to March 2017). The longitudinal study was designed to evaluate the dietary fiber intake over the 1-year observation between the first and the second follow-up on gut microbiota at the second follow-up. A total of 356 participants from the first follow-up survey, 310 participants from the second survey, and 293 patients participating in both surveys were included in this study.

### 2.2. Data Collection

In the parent RCT, in-person interviews were conducted at baseline by trained community healthcare staff using a structured questionnaire to collect information on demographic characteristics, lifestyle factors, diagnosis of T2DM, and the basic health status of T2DM patients. During the first and the second follow-up surveys, a 3-day 24-h dietary recall was conducted for each subject to collect the date of their record, their mealtime, the amount of food they consumed, and recipes containing many ingredients. Energy intake and insoluble fiber intake were calculated based on the estimated data on habitual diets and the *Chinese Food Composition Table (2nd Edition*) [21]. No subject reported to take any nutritional supplements. To evaluate the 1-year dietary fiber intake between the first and the second follow-up, all subjects were classified as lower-lower, higher-lower, lower-higher, or higher-higher groups using a cut-off value of 7.2 g/day according to their fiber intake during the two surveys. The groups of lower-lower and higher-higher were considered to have a stable dietary fiber intake.

Physical activities were collected using the Global Physical Activity Questionnaire (GPAQ) (https://www.who.int/ncds/surveillance/steps/GPAQ_CH.pdf, accessed date: 25 February 2022) and quantified as metabolic equivalent (MET)-hours/week [22,23]. Use of anti-diabetes agents during the period was obtained by a record linkage with a local diabetes management system.

At each survey, the body weight, standing height, circumferences of waist and hip, blood pressure, and heart rate of each subject was measured following a standardized protocol. Body mass index (BMI) and waist-to-hip ratio (WHR) were calculated using the measured values. HbA1c level was assayed at qualified Community Healthcare Centers (CHCs) using the high-performance liquid chromatography according to the recommendation of the National Glycohemoglobin Standardization Program.

### 2.3. Stool Sample Collection

At the first and second follow-up surveys, each subject was instructed to collect a peanut-sized stool sample using a tube containing 5 mL 95% ethanol and a couple of glass beads. All samples were mixed immediately by shaking the tubes up and down for 10 times or more, and then delivered to the CHCs and the lab in Fudan University within 24 h. After centrifugalization, each sample was divided into two vials, and stored separately at −80 °C immediately for DNA extraction and sequencing.

Using a brief structured questionnaire, all subjects were asked to record the date and time of sample collection, and self-report the usage of antibiotics, nonsteroidal anti-inflammatory drugs (NSAIDs), and other medications within 1 week and six months before stool donation. The samples collected from those using antibiotics were excluded from further assays.

### 2.4. Bacterial DNA Sequencing and Bio-Informatics Analysis

Fecal microbiota DNA were extracted from stool samples using the fecal genome extraction kit (Ace111, AcebioX^®^) following the manufacturer’s instructions. PCR amplification of 16S rDNA V4 region was performed using Roche Lightcycler 480II and the following primers: 5′ GAGTGCCAGCMGCCGCGGTAA 3′ and 5′ ACGGACTACHVGGGTWTCTAAT 3′ (AcebioX qPCR Mastermix Ace103). The targeted fragments were recycled with a DNA gel extraction kit (CW2302S, CWBIO^®^) for the DNA library. DNA sequencing was performed based on the Illumina Hiseq platform at the AcebioX company in Shanghai, China.

16S rDNA sequenced data were analyzed using QIIME 2 [24]. After removing the primer, stitching paired-end reads and filtering, the raw reads from Illumina were processed to clean reads. Low-quality sequences and chimeras were eliminated for the construction of Operational Taxonomic Units (OTUs) and taxa assignment. The relative abundance of each assigned taxa was calculated for statistical analyses. α-diversity was measured using Chao1 and the Shannon index based on 61,608 randomly selected reads from each sample (http://scikit-bio.org/docs/latest/generated/skbio.diversity.alpha.html, accessed date: 30 October 2018). The good’s coverages were all greater than 99.8%. β-diversity was evaluated based on the Bray—Curtis distance between samples.

### 2.5. Statistical Analysis

Continuous variables were presented as the median and interquartile range (IQR), and categorical variables were presented as counts and percentages.

A generalized linear model (GLM) was applied to evaluate the associations of α-diversity of gut microbiota with dietary fiber intake and HbA1c level. A restricted cubic spline (RCS) curve was drawn to illustrate the changing pattern of α-diversity along with dietary fiber intake, and thus identify potential cut-off values. Piecewise regression was used to conduct sensitivity analysis by using different cut-off values of dietary fiber.

For β-diversity, a principal co-ordinates analysis (PCA) was performed to reduce the dimension of distance matrix, and then a permutational multivariate analysis of variance (PERMANOVA) was conducted to compare the differences in Bray—Curtis distances by dietary fiber intake and HbA1c level after adjusting for covariates.

For individual taxa, analyses were carried out at the phylum and genus level and those presented in <10% of participants were excluded from the analyses. Microbiome Multivariable Associations with Linear Models (MaAsLin) were used to examine the associations of arcsin-square root transformed abundance of related taxa [25] with dietary fiber intake or HbA1c level, with a Benjamini—Hochberg (BH) false discovery rates (FDRs) corrected *Q*-value calculated for *p*-values at each taxonomic level. *Q*-value < 0.15 was considered significant due to the small sample size in this study.

Mediation analysis was conducted to fit a mediator model for *M* (gut microbiota) and *X* (fiber intake) and an outcome model for *Y* (HbA1c level), respectively, and then integrated into one mediation model using GLM or logistic regression [26]. Nonparametric bootstrap was used to estimate the confidence intervals (CIs) of effects with 1000 resamples [26].

Longitudinal analyses were conducted by using dietary fiber intake over the 1-year observation as an exposure, while gut microbiota or HbA1c level at the second follow-up as the outcomes. Age, sex, educational level, duration of diabetes, intervention status in parent trial, physical activity level, total energy intake, use of anti-diabetes agents and NSAIDs within six months before stool collection were adjusted in all models as covariates. All statistical analyses were performed using R version 4.1.0. *p*-value < 0.05 was considered statistically significant.

## 3. Results

### 3.1. Characteristics of Participants of the Baseline, First and Second Follow-Up Surveys

As shown in Table 1, the median intake of dietary fiber was 6.9 g/day (IQR: 4.6 to 10.0) at baseline, slightly lower than 7.4 g/day (IQR: 5.5 to 9.7) at the first follow-up and 7.1 g/day (IQR: 5.1 to 11.1) at the second follow-up. HbA1c level was observed higher in the second than in the first follow-up and at baseline, with respective medians of 8.0%, 7.8% and 7.8%. The glycemic control rate was 15.8% at baseline, 26.4% at the first follow-up and 23.9% at the second follow-up. Considering the limited number of patients with controlled glycemic status, we further classified all subjects using a cut-off value of 8.0% of HbA1c, which is a suggested control goal for severe diabetes patients according to the *Guidelines for the prevention and control of type 2 diabetes in China (2017 Edition)* [27]. No significant difference was observed for the rest factors between the two follow-up surveys. A total of 31 participants from the first follow-up survey and 9 from the second follow-up survey reported using antibiotics within 6 months of stool sample collection.

### 3.2. Associations of Habitual Dietary Fiber Intake with HbA1c Level

As shown in Appendix A, no significant change in HbA1c level was observed along with increasing dietary fiber intake after adjusting for potential confounders in cross-sectional and longitudinal analyses. Further analysis using glycemic control status (defined using a cut-off point of 7.0% or 8.0%) as a dependent variable did not find any significant association with dietary fiber intake either (Appendix A).

### 3.3. Diversity of Fecal Microbiota by Habitual Dietary Fiber Intake and HbA1c Level

After excluding 40 antibiotics users, a total of 325 participants from the first follow-up and 301 subjects from the second follow-up were included in the analyses. As a result, 284 participants from the two surveys were used to evaluate the associations of dietary fiber intake over the 1-year observation with fecal microbiota profiles.

To identify the optimal cut-off value of dietary fiber intake, we drew an RCS curve for α-diversity along with dietary fiber intake, and observed obvious segmented trends of Chao1 and the Shannon index along with fiber intake at the first follow-up (Appendix A). Further piecewise regression identified an optimal cut-off point at 7.2 g/day of dietary fiber intake (Appendix A).

Table 2 shows the negative associations of dietary fiber intake with α-diversity of gut microbiota in our subjects. The associations were significant only among patients taking dietary fiber ≥ 7.2 g/day in the first follow-up, with a regression coefficient β [95%CI] of −0.06 (−0.11, −0.01) for the Shannon index. In the second follow-up, the significant negative association with Chao1 in all subjects (β: −2.06; 95%CI: −3.66, −0.46) did not remain significant within subjects taking dietary fiber <7.2 g/d or ≥7.2 g/d, but for the higher versus the lower group β: −28.78; 95%CI: −47.58, −9.97. In the longitudinal analysis, the higher-higher group had less Chao1 than the lower-lower group, with β [95%CI] of −33.63 [−59.03, −8.23]. For β-diversity, a significant difference was observed for the Bray—Curtis distance between subgroups taking higher and lower fiber at the second follow-up (*p* = 0.014), but not at the first follow-up (Appendix A).

We observed a positive association of Chao1 [β: 0.00%; 95%CI: 0.00, 0.00; *p* = 0.021] and the Shannon index [β: 0.27%; 95%CI: 0.08, 0.46; *p* = 0.005] with HbA1c in participants from the first follow-up, but not among those from the second follow-up. No significant difference in β-diversity was observed by HbA1c level either.

### 3.4. Differential Taxa by Glycemic Control Status and Habitual Dietary Fiber Intake

As shown in Table 3, phylum *Euryarchaeota*, genus *Methanobrevibacter*, *Prevotellaceae (Unknown)*, *Ruminococcaceae (Unknown)*, *Holdemania*, *Methylobacteriaceae (Unknown)*, and *Desulfovibrio* were positively associated with HbA1c level (*p* < 0.05, *Q* < 0.15) at the first follow-up. Genus *Desulfovibrio* and *Methylobacteriaceae (Unknown)* were also more abundant in participants from the first follow-up with HbA1c ≥ 8.0% (*p* < 0.05, *Q* < 0.15) (Appendix A). At the second follow-up, only genus *Epulopiscium (p* = 0.001, *Q* = 0.122) were more enriched in glycemic controlled patients (Appendix A).

Differential taxa by habitual dietary fiber intake are presented in Table 4. In participants from the first follow-up, the relative abundance of phylum *Firmicutes* was negatively associated with fiber intake among subjects taking dietary fiber ≥ 7.2 g/day (*p* < 0.05, *Q* < 0.15). Phylum *Fusobacteria* was also less enriched in those taking fiber ≥ 7.2 g/day than those taking less (*p* < 0.05, *Q* < 0.15). In the second follow-up, compared with patients taking dietary fiber < 7.2 g/day, those taking fiber ≥ 7.2 g/day had less abundant genus *Adlercreutzia*, *Prevotella*, *Ruminococcus*, and *Desulfovibrio (p* < 0.05, *Q* < 0.15). No differential taxa were found between the groups with higher-higher and lower-lower dietary fiber intake over the 1-year observation in the longitudinal analysis.

Of the taxa slightly associated with dietary fiber intake, genus *Desulfovibrio* and *Ruminococcaceae (Unknown)*, two bacteria positively associated with HbA1c level, were consistently found to be less abundant among subjects taking higher dietary fiber in the two cross-sectional and the longitudinal analyses.

### 3.5. Mediation Effect of Gut Microbiota in Dietary Fiber-HbA1c Associations

Although α-diversity was associated with dietary fiber intake and HbA1c level among patients taking higher fiber at the first follow-up, we did not observe a significant mediation effect of the Shannon index in dietary fiber-HbA1c associations (Figure 2A). On the other hand, genus *Desulfovibrio* and *Ruminococcaceae (Unknown)*, the two common differential taxa for dietary fiber intake and HbA1c level among patients taking dietary fiber ≥ 7.2 g/day at the first follow-up, were observed to mediate the dietary fiber-HbA1c association negatively, with β [95%CI] being −0.019 [−0.043, −0.003] (*p* = 0.018) and −0.019 [−0.046, −0.003] (*p* = 0.016), respectively (Figure 2B,C).

## 4. Discussion

In this study based on a parent cluster RCT in Chinese diabetes patients, we did not observe a significant association between habitual dietary fiber intake and HbA1c level. However, we found a negative association between dietary fiber intake and α-diversity estimated by Chao1 and the Shannon index, particularly among patients taking dietary fiber ≥ 7.2 g/day. We also observed a higher relative abundance of genus *Bacteroides*, and a lower relative abundance of phylum *Firmicutes* and *Fusobacteria*, genus *Adlercreutzia*, *Prevotella*, *Ruminococcaceae (Unknown)*, *Ruminococcus*, and *Desulfovibrio* among patients taking higher dietary fiber, in which *Desulfovibrio* and *Ruminococcaceae (Unknown)* were observed to mediate the dietary fiber-HbA1c association. Our results suggest that diabetes patients with low to moderate fiber intakes may have some beneficial changes in gut microbiota; however, these changes are not robust enough to elicit clinical improvements.

The negative association between habitual dietary fiber and α-diversity was contrary to most previous findings [28,29,30], particularly in intervention trials using high-fiber diets [15]. The inconsistency may be explained by the low level of dietary fiber intake in our subjects, which was only 7.4 g/day on average at the first follow-up and 7.1 g/day at the second follow-up, far below the average level in adults in China (17~19 g/day) [31] and the world [19], as well as the recommended level for diabetes patients (10~14 g/1000 kcal·day) [27]. Low-level fiber could induce a shift in gut microbial metabolism towards the utilization of proteins, fat, lactic acid, and other nutrients [32], and consequently reduce the production of SCFAs [33]. The altered gut microbiota in diabetes patients were dominated by multiple harmful bacteria [34,35], which were also associated with poor glycemic response to low-fiber diets [33]. In fact, the association between dietary fiber intake and the risk of T2DM has been demonstrated in a dose-response pattern [36]. These results suggest that a relatively higher fiber intake in habitual diet might lead to favorable alternations in gut microbiota by restricting the growth of harmful bacteria, even if the effect is not enough to improve HbA1c level.

Our hypothesis is supported by the fact that most bacterial taxa negatively associated with dietary fiber intake in our subjects were reported detrimental in previous studies. Specifically, genus *Desulfovibrio*, a bacterium producing endotoxin [37] which is prone to being promoted by a high-fat diet [38], was found to cause abnormal metabolism [39]. Genus *Ruminococcus*, *Ruminococcaceae (Unknown)*, and *Megamonas*, taxa belonging to family *Ruminococcaceae*, were found to be enriched in hyperglycemic mice [40]. Genus *Ruminococcus* was observed to be less abundant in humans taking rich-vegetable diets [41], and was consistently associated with an increased risk of T2DM [34]. Genus *Adlercreutzia* was less abundant in gestational hypertensive women taking higher dietary fiber [42], but more prevalent in healthy controls than patients with diabetes [43] and gestational diabetes [44]. Genus *Prevotella* was also a detrimental bacterium linked to chronic systemic inflammation [45] and dominated dysbiosis in T2DM patients [46]. However, in Wu’s study [47], *Prevotella* was more abundant in participants with a high fiber diet, which is consistent with its capability to ferment non-digestible polysaccharides. Genus *Bacteroides*, the only taxa positively associated with dietary fiber intake, was able to produce corresponding enzymes and degrade non-digestible carbohydrates [48].

Of the eight taxa related with HbA1c level in this study, genus *Desulfovibrio* was associated with T2DM through the Mendelian randomization approach [49], while *Ruminococcaceae* was widely reported to be associated with T2DM [34,40], and *Holdemania* was related with metabolic disorders in the elderly [50]. Our results, consistent with previous studies or not, indicate the importance of gut microbiota in glycemic control among Chinese T2DM patients. It is of note that in this study, most differential taxa by HbA1c level were not related with dietary fiber intake, suggesting that the compositions of fecal microbiota might be induced by other factors rather than dietary fiber.

As the common differential taxa by HbA1c level and dietary fiber intake, genus *Desulfovibrio* and *Ruminococcaceae (Unknown)* were found to mediate the limited effect of dietary fiber on HbA1c level. It seems that, once dietary fiber intake reached a threshold (≥7.2g/d in our subjects), higher fiber intake may help to decrease the abundance of harmful bacterial genera, and thus improve HbA1c level. So far, few previous studies have quantified the mediation effect of gut microbiota between diets and human health. Menni et al. [51] found that *Collinsella* mediated 20% of the effect of high vegetable intake on lower inflammatory lymphocyte counts, supporting our hypothesis that higher dietary fiber suppresses the growth of harmful bacteria in gut.

In this study, we did not identify shared bacterial taxa associated with dietary fiber intake in the first and second follow-up survey, which may present different gut microbiota profiles caused by active interventions (i.e., at the first follow-up) and ceased interventions (i.e., at the second follow-up).

To our knowledge, this is the first study to evaluate the effect of habitual dietary fiber intake on gut microbiota and glycemic control in diabetes patients. The main strength of the study is the cross-sectional and longitudinal analyses within a well-designed cluster RCT. The detailed baseline and follow-up data make it possible to evaluate the associations comprehensively and adjust for various covariates. However, several limitations of the study should be mentioned. First, our subjects were diabetes patients with poor glycemic control status who would have taken multiple medications. Considering the small sample size, we did not exclude patients using anti-diabetic agents, NSAIDs or other drugs. Residual confounding effects may exist due to adjusting for use of these medications. Second, the small sample size also limited the statistical power of the analyses. Therefore, we focused on the taxa slightly associated with dietary fiber intake (e.g., *p*-value < 0.05 but *Q*-value > 0.15) to avoid possible type I error. Third, we could only estimate the amount of insoluble fiber intake according to the *China Table of Food Composition (2nd Edition*), but we neglected soluble fiber which may also elicit effects on human health [52,53]. The impact of the few soluble dietary fiber in diets consumed by the Chinese patients [19], if any, may be small and can be ignored. The low dietary fiber intake also limited our effective analysis by fiber sources. Finally, loss to follow-up may cause selection bias, and the 1-year interval between the first and the second follow-up survey may have caused information bias due to possible changed lifestyle and health status during the long period. However, no significant difference in baseline characteristics was observed between participants and nonparticipants of both the first and the second follow-up, partly releasing our concern on selection bias. Furthermore, since no intervention was conducted to change lifestyle in our subjects between the two surveys, we did not find any significant changes in the characteristics and lifestyle factors among the 293 subjects participating in both surveys, indicating the stable lifestyles of our subjects over the 1-year period.

## 5. Conclusions

In summary, we observed a beneficial effect of habitual dietary fiber on gut microbiota in Chinese diabetes patients who usually consume refined grain habitual diets. The effect is probably achieved by decreasing harmful bacteria in the gut, but the findings do not appear robust enough to improve HbA1c level. A high fiber diet should be advocated in populations for better clinical outcomes.

## Figures and Tables

**Figure 1 nutrients-14-01003-f001:**
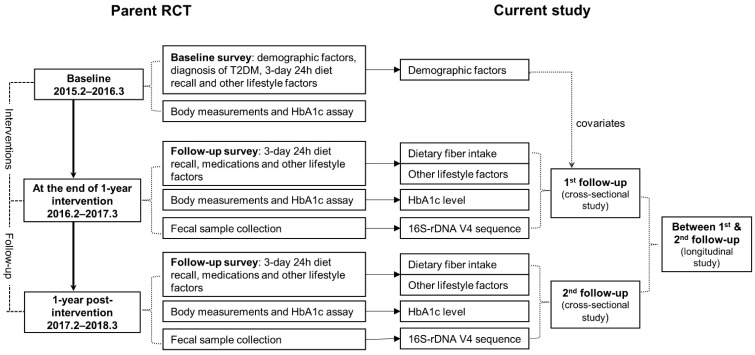
Diagram of study design.

**Figure 2 nutrients-14-01003-f002:**
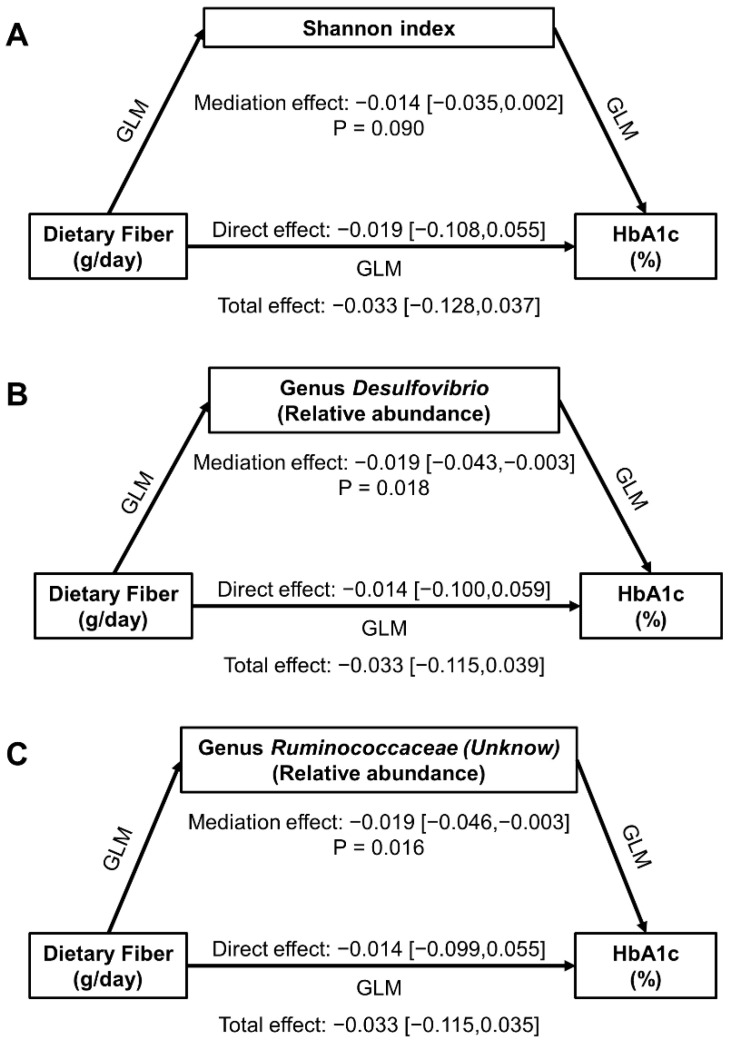
Mediation effect of fecal microbiota in associations of dietary fiber intake with HbA1c level at the first follow-up among 172 participants taking dietary fiber ≥ 7.2 g/day. (**A**) for Shannon index, (**B**) for genus *Desulfovibrio*, and (**C**) for genus *Ruminococcaceae (Unknown)*. Effect presented as β [95% CI], and adjusted for age, sex, education level, duration of diabetes, intervention status in parent trial, physical activity level, energy intake, fat intake, use of anti-diabetes agents and NSAIDs within six months before stool collection in all models.

**Table 1 nutrients-14-01003-t001:** Characteristics of participants of the parent RCT at baseline and follow-up surveys.

Characteristics	At Baseline (*n* = 400)	At the First Follow-Up (*n* = 356)	At the Second Follow-Up (*n* = 310)
Age at survey (years)	65.0 (58.0, 70.0)	66.5 (60.0, 71.5)	68.0 (60.0, 73.0)
Sex			
Men	174 (43.5)	155 (43.5)	132 (42.6)
Women	226 (56.5)	201 (56.5)	178 (57.4)
Educational level			
Primary school or below	96 (24.0)	88 (24.7)	77 (24.8)
Junior high school	146 (36.5)	130 (36.5)	114 (36.8)
Senior high or vocational school	97 (24.2)	81 (22.8)	71 (22.9)
College or above	61 (15.3)	57 (16.0)	48 (15.5)
Duration of diabetes (year)	9 (5, 14)	10 (6, 15)	11 (7, 17)
Diagnosis of hypertension	272 (68.0)	244 (68.5)	213 (68.7)
Diagnosis of hyperlipidemia	126 (31.5)	117 (32.9)	97 (31.3)
Intervention status in parent trial			
Health literacy group	100 (25.0)	84 (23.6)	75 (24.2)
Exercise Group	100 (25.0)	97 (27.3)	77 (24.8)
Comprehensive group	100 (25.0)	92 (25.8)	88 (28.4)
Control group	100 (25.0)	83 (23.3)	70 (22.6)
Body mass index (kg/m^2^)	24.9 (23.1, 26.8)	24.5 (22.9, 26.5)	24.9 (23.0, 26.6)
Waist-to-hip ratio	0.90 (0.87, 0.93)	0.90 (0.87, 0.92)	0.90 (0.87, 0.93)
Physical activity (METs-hours/week)	61 (37, 91)	60 (34, 88)	56 (29, 98)
Use of anti-diabetes medicines			
Never	12 (3.0)	11 (3.1)	11 (3.6)
Insulin shot only	41 (10.3)	33 (9.3)	33 (10.7)
Oral agents only	309 (77.3)	251 (70.5)	215 (69.4)
Both	38 (9.5)	61 (17.1)	51 (16.5)
Total energy intake (kcal/day)	1557 (1211, 1864)	1570 (1261, 1899)	1643 (1310, 2033)
Sources of calorie (%)			
Carbohydrate	39 (33, 44)	36 (30, 41)	33 (29, 39)
Fat	45 (40, 51)	48 (41, 53)	50 (44, 55)
Protein	15 (13, 17)	16 (14, 18)	16 (13, 17)
Dietary fiber intake (g/d)	6.9 (4.5, 10.0)	7.4 (5.5, 9.7)	7.1 (5.1, 11.1)
HbA1c level (%)	7.8 (7.2, 8.6)	7.8 (6.9, 9.0)	8.0 (7.0, 9.2)
Glycemic status: HbA1c < 7.0%	63 (15.8)	94 (26.4)	74 (23.9)
Glycemic status: HbA1c < 8.0%	220 (55.0)	189 (53.1)	153(49.4)
Medications at sample collection			
Use of antibiotics	--	31 (8.7)	9 (2.9)
Use of NSAIDs	--	69 (19.4)	46 (14.8)

Data are presented as median (IQR) for continuous variables or count (percentage) for categorical variables.

**Table 2 nutrients-14-01003-t002:** Associations of dietary fiber intake with α-diversity of gut microbiota in Chinese diabetes patients.

	No. of Subjects	Chao1			Shannon		
		Median (IQR)	β [95%CI]	*p*-Value	Median (IQR)	β [95%CI]	*p*-Value
At the first follow-up							
All subjects	325	425 (379, 488)	0.63 [−2.12, 3.37]	0.654	4.36 (3.68, 4.95)	−0.02 [−0.05, 0.01]	0.146
By dietary fiber intake (g/day)							
Lower (<7.2)	153	425 (379, 480)	−0.83 [−11.46, 9.81]	0.878	4.35 (3.68, 5.00)	−0.07 [−0.19, 0.06]	0.299
Higher (≥7.2)	172	425 (379, 491)	−4.11 [−8.68, 0.47]	0.078	4.37 (3.68, 4.89)	−0.06 [−0.11, −0.01]	0.011
Higher versus lower			14.98 [−4.16, 34.11]	0.125		0.00 [−0.20, 0.21]	0.964
At the second follow-up							
All subjects	301	406 (358, 462)	−2.06 [−3.66, −0.46]	0.012	4.20 (3.74,4.77)	−0.01 [−0.02, 0.01]	0.427
By dietary fiber intake (g/day)							
Lower (<7.2)	152	424 (362, 491)	−1.68 [−12.65, 9.29]	0.763	4.27 (3.65, 4.89)	−0.05 [−0.15, 0.06]	0.352
Higher (≥7.2)	149	390 (358, 433)	−0.96 [−2.77, 0.85]	0.297	4.17 (3.80, 4.57)	−0.01 [−0.03, 0.01]	0.419
Higher versus lower			−28.78 [−47.58, −9.97]	0.003		0.00 [−0.19, 0.19]	0.979
Longitudinal analysis							
Lower-lower	80	424 (364, 490)	ref		4.31 (3.66, 4.89)	ref	
Higher-lower	62	428 (381, 544)	14.92 [−11.26, 41.11]	0.263	4.28 (3.70, 4.95)	−0.04 [−0.30, 0.23]	0.773
Lower-higher	60	390 (358, 429)	−24.50 [−51.06, 2.07]	0.071	4.09 (3.63, 4.58)	−0.10 [−0.37, 0.17]	0.450
Higher-higher	82	391 (358, 435)	−33.63 [−59.03, −8.23]	0.010	4.15 (3.85, 4.57)	−0.15 [−0.41, 0.11]	0.252

β coefficients [95%CI] for α-diversity with per unit dietary fiber intake (g/day) or with the higher (≥7.2 g/day) versus the lower dietary fiber intake (<7.2 g/day), and adjusted for age, sex, educational level, duration of diabetes, intervention status in parent trial, physical activity level, energy intake, fat intake, use of anti-diabetes agents and NSAIDs within six months before stool collection.

**Table 3 nutrients-14-01003-t003:** Linear associations of relative abundances of taxa with HbA1c level in Chinese diabetes patients.

Phylum	Class	Order	Family	Genus/Species	Relative Abundance (%)	Transformed Abundance (%)	β [95%CI] for HbA1c (%)	*p*-Value	*Q*-Value
All subjects at the first follow-up (*n* = 325)									
*Euryarchaeota*					0.000 (0.000, 0.003)	0.000 (0.000, 0.506)	0.19 [0.08, 0.31]	0.001	0.013
*Euryarchaeota*	*Methanobacteria*	*Methanobacteriales*	*Methanobacteriaceae*	*Methanobrevibacter*	0.000 (0.000, 0.002)	0.000 (0.000, 0.438)	0.24 [0.10, 0.38]	0.001	0.053
*Bacteroidetes*	*Bacteroidia*	*Bacteroidales*	*Prevotellaceae*	*(Unknown)*	0.000 (0.000, 0.001)	0.000 (0.000, 0.386)	0.30 [0.09, 0.52]	0.006	0.131
*Firmicutes*	*Clostridia*	*Clostridiales*	*Ruminococcaceae*	*(Unknown)*	0.743 (0.243, 2.283)	8.630 (4.928, 15.168)	0.03 [0.01, 0.06]	0.006	0.131
*Firmicutes*	*Erysipelotrichi*	*Erysipelotrichales*	*Erysipelotrichaceae*	*Holdemania*	0.006 (0.002,0.014)	0.754 (0.390, 1.184)	0.33 [0.11, 0.55]	0.004	0.120
*Proteobacteria*	*Alphaproteobacteria*	*Rhizobiales*	*Methylobacteriaceae*	*(Unknown)*	0.000 (0.000, 0.000)	0.000 (0.000,0.000)	1.66 [0.88, 2.44]	0.000	0.006
*Proteobacteria*	*Deltaproteobacteria*	*Desulfovibrionales*	*Desulfovibrionaceae*	*Desulfovibrio*	0.036 (0.022, 0.191)	1.909 (1.472, 4.367)	0.05 [0.02, 0.09]	0.001	0.053
All subjects at the second follow-up (*n* = 301)									
*--*									

All taxa listed with *Q*-values less than 0.15; abundance of taxa presented as median (IQR); β coefficients [95%CI] for HbA1c with per unit of transformed abundance of taxa, and adjusted for age, sex, education level, duration of diabetes, intervention status in parent trial, physical activity level, energy intake, fat intake, use of anti-diabetes agents and NSAIDs within six months before stool collection.

**Table 4 nutrients-14-01003-t004:** Associations of habitual dietary fiber intake with relative abundances of taxa in Chinese diabetes patients.

Phylum	Class	Order	Family	Genus	Relative Abundance (%)	Transformed Abundance (%)	β [95%CI] with Dietary Fiber Intake	*p*-Value	*Q*-Value
At the first follow-up									
All subjects (*n* = 325)									
*Firmicutes*	*Clostridia*	*Clostridiales*	*Ruminococcaceae*	*Megamonas*	0.341 (0.229, 1.295)	5.847 (4.782, 11.405)	−0.46 [−0.89, −0.03]	0.038	0.745
Subjects taking dietary fiber ≥ 7.2 g/day (*n* = 172)									
*Firmicutes*					35.830 (24.757, 43.854)	64.173 (52.077, 72.378)	−0.98 [−1.78, −0.18]	0.017	0.110
*Bacteroidetes*	*Bacteroidia*	*Bacteroidales*	*Bacteroidaceae*	*Bacteroides*	19.104 (9.551, 35.675)	45.235 (31.419, 64.011)	1.36 [0.26, 2.46]	0.017	0.393
*Firmicutes*	*Clostridia*	*Clostridiales*	*Ruminococcaceae*	*(Unknown)*	0.738 (0.279, 2.318)	8.599 (5.281, 15.283)	−0.43 [−0.79, −0.06]	0.023	0.441
*Proteobacteria*	*Deltaproteobacteria*	*Desulfovibrionales*	*Desulfovibrionaceae*	*Desulfovibrio*	0.037 (0.021, 0.202)	1.928 (1.433, 4.492)	−0.32 [−0.59, −0.04]	0.024	0.441
Dietary fiber intake: ≥7.2 g/day (*n* = 172) versus <7.2 g/day (*n* = 153)									
*Fusobacteria*					0.162 (0.097, 0.405)	4.026 (3.111, 6.368)	−3.27 [−5.68, −0.87]	0.008	0.103
At the second follow-up									
All subjects (*n* = 301)									
*Firmicutes*	*Clostridia*	*Clostridiales*	*Ruminococcaceae*	*(Unknown)*	0.644 (0.276, 1.780)	8.035 (5.257, 13.381)	−0.15 [−0.27, −0.03]	0.014	0.448
Dietary fiber intake: ≥7.2 g/day (*n* = 149) versus <7.2 g/day (*n* = 152)									
*Actinobacteria*	*Coriobacteriia*	*Coriobacteriales*	*Coriobacteriaceae*	*Adlercreutzia*	0.001 (0.000, 0.006)	0.354 (0.000, 0.774)	−0.23 [−0.39, −0.08]	0.003	0.139
*Bacteroidetes*	*Bacteroidia*	*Bacteroidales*	*Bacteroidaceae*	*Bacteroides*	25.017 (8.147, 41.863)	52.380 (28.945, 70.367)	7.19 [1.43, 12.94]	0.015	0.249
*Bacteroidetes*	*Bacteroidia*	*Bacteroidales*	*Prevotellaceae*	*Prevotella*	2.865 (1.857, 34.708)	17.008 (13.671, 62.999)	−10.95 [−18.40, −3.50]	0.004	0.139
*Firmicutes*	*Clostridia*	*Clostridiales*	*Ruminococcaceae*	*Ruminococcus*	0.272 (0.108, 0.905)	5.215 (3.287, 9.525)	−2.05 [−3.32, −0.78]	0.002	0.139
*Firmicutes*	*Clostridia*	*Clostridiales*	*Ruminococcaceae*	*(Unknown)*	0.644 (0.276, 1.780)	8.035 (5.257, 13.381)	−1.43 [−2.86, −0.01]	0.050	0.330
*Proteobacteria*	*Deltaproteobacteria*	*Desulfovibrionales*	*Desulfovibrionaceae*	*Desulfovibrio*	0.041 (0.021, 0.188)	2.019 (1.441, 4.341)	−1.81 [−3.07, −0.54]	0.005	0.139
Longitudinal analysis									
Higher-higher (*n* = 82) versus Lower-lower (*n* = 80)									
*Firmicutes*	*Clostridia*	*Clostridiales*	*Ruminococcaceae*	*Megamonas*	0.354 (0.237, 0.870)	5.953 (4.874, 9.340)	−4.42 [−8.33, −0.51]	0.027	0.347
*Proteobacteria*	*Deltaproteobacteria*	*Desulfovibrionales*	*Desulfovibrionaceae*	*Desulfovibrio*	0.041 (0.021, 0.198)	2.037 (1.454, 4.455)	−2.10 [−3.71, −0.49]	0.011	0.319

All taxa listed with *p*-values less than 0.05; abundance of taxa presented as median (IQR); β coefficients [95%CI] for transformed abundance of taxa with per unit dietary fiber intake (g/day) or as the higher (≥7.2 g/day) versus the lower dietary fiber intake (<7.2 g/day), and adjusted for age, sex, education level, duration of diabetes, intervention status in parent trial, physical activity level, energy intake, fat intake, use of anti-diabetes agents and NSAIDs within six months before stool collection.

## Data Availability

Data described in the article, code book, and analytic codes will be made available upon request.

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
