# Peer review of "Habitual Dietary Fiber Intake, Fecal Microbiota, and Hemoglobin A1c Level in Chinese Patients with Type 2 Diabetes"

_nutrients, 2022, doi:10.3390/nu14051003_

Round 1

Reviewer 1 Report

The paper “Habitual dietary fiber intake, fecal microbiota, and hemoglobin A1c level in Chinese patients with type 2 diabetes" by Fu et al. is a randomized control trial with the aim of assessing of the potential beneficial effects of habitual dietary fiber intake.

The article is well written. The study has a good design. The article is logically divided into sections and subsections. The references cited are relevant and adequate. The work has an average degree of novelty and of good interest to the readers.

Comments:

  • Introduction should be extended providing more information about microbiota and its role an difference. In particular, line 41 should focus on providing details about the complexity of the gut microbiota “Human microbiota has been estimated to include about 10–100 trillion microbes, mostly located in the large intestine, for a maximum weight of 1.5 Kg and more than 1000 bacterial type” (DOI: 10.3390/antiox10020270). Line 43 should underline another important issue, represented by the different types of gut microbiota to tie up with the rest of the paper: “In fact, there is a microbiota defined as “obese”, i.e., with a higher ability to absorb energy from the diet, seems able to determine a significantly higher increase in total body fat as compared to an individual colonized by the so-called “lean” microbiota (DOI: 3390/pr9010135). Line 52: “gut microbiota-derived lipopolysaccharide levels are increased exhibit endotoxic effects and can be responsible of non-alcoholic fatty liver onset and progression, as well as increased cardiovascular disease (DOI: 10.31083/J.RCM2203082).
  • Line 78-79: why using the 1999 WHO criteria for diabetes diagnosis? To my knowledge, WHO 2019 are the most recent WHO guidelines, or, even better, American Diabetes Association 2022 are the most used and updated.
  • Baseline patients are 400, at 1st follow up 356 (missing 11%) and 2nd follow up 310 (22.5%), this missing may cause an important bias which should be added to the limitations of the RCT.

Reviewer 2 Report

Increased fiber in everyday diet was reported to decrease HbA1c, improve lipid profile and blood glucose levels in diabetic patients. In the manuscript, the authors collected samples from about 300 patients in each follow-up to test the association of fiber intake, microbiome and HbA1C levels. The manuscript is interesting. Here are two questions.

  1. Since the authors focused on habitual dietary fiber, what is the other information about these patients’ diets. For example, how much calories did they have? What is the component of their diets? Different fat and sugar levels would affect the conclusions.
  2. As the authors discussed in the manuscript, the cut-off level of fiber intake is lower than the average level in China at the same time period. Though if increase the cut-off, the authors might have fewer patient numbers, it will be more convincing to also add the data from the patients whose fiber intake was at least equal to the average level and compare the microbiome and HbA1C.

Round 2

Reviewer 1 Report

I have no more comments to the paper.